# Global Cardiovascular Risk Profile of Italian Medical Students Assessed by a QR Code Survey. Data from UNIMI HEART SURVEY: Does Studying Medicine Hurt?

**DOI:** 10.3390/jcm10071343

**Published:** 2021-03-24

**Authors:** Andrea Faggiano, Francesca Bursi, Gloria Santangelo, Cesare Tomasi, Chiarella Sforza, Pompilio Faggiano, Stefano Carugo

**Affiliations:** 1Fondazione IRCCS Ca’ Granda Ospedale Maggiore Policlinico, Cardiovascular Disease Unit, Internal Medicine Department, University of Milan, 20122 Milan, Italy; stefano.carugo@unimi.it; 2Division of Cardiology, Heart and Lung Department, San Paolo Hospital, ASST Santi Paolo and Carlo, University of Milan, 20142 Milano, Italy; francescabursi@gmail.com (F.B.); gloriasantangelo1982@gmail.com (G.S.); 3Section of Occupational Medicine and Industrial Hygiene, University of Brescia, 25133 Brescia, Italy; cesare.tomasi@live.com; 4Dipartimento di Scienze Biomediche per la Salute, University of Milan, 20122 Milan, Italy; chiarella.sforza@unimi.it; 5Fondazione Poliambulanza, Cardiovascular Disease Unit, University of Brescia, 25124 Brescia, Italy; cardiologia@pompiliofaggiano.it

**Keywords:** QR code, survey, primary prevention, cardiovascular risk factors, young adult, medical students

## Abstract

Background: Few studies to date have addressed global cardiovascular (CV) risk profile in a “protected” young population as that of medical school students. Objective: to assess CV traditional risk factors and global CV risk profile of Italian medical students throughout the six years of university. Methods: A cross-sectional survey accessible online via quick response (QR) code was conducted among 2700 medical students at the University of Milan, Italy. Data on baseline characteristics, traditional CV risk factors, diet, lifestyle habits, and perceived lifestyle variations were evaluated across different years of school. Results: Overall, 1183 students (mean age, 22.05 years; 729 women (61.6%)) out of 2700 completed the questionnaire (43.8% rate response). More than 16% of the students had at least 3 out of 12 CV risk factors and only 4.6% had ideal cardiovascular health as defined by the American Heart Association. Overweight, underweight, physical inactivity, sub-optimal diet, smoke history, and elevated stress were commonly reported. Awareness of own blood pressure and lipid profile increased over the academic years as well as the number of high-blood-pressure subjects, alcohol abusers, and students constantly stressed for university reasons. Moreover, a reduction in physical-activity levels over the years was reported by half of the students. Conclusion and Relevance: This study demonstrates that a “protected” population as that of young medical students can show an unsatisfactory cardiovascular risk profile and suggests that medical school itself, being demanding and stressful, may have a role in worsening of the lifestyle.

## 1. Introduction

Current guideline-suggested assessment of cardiovascular (CV) risk profile, based on individual 10-year predicted risk of a CV event, is mainly influenced by age and thus appears to be unnecessary among young people [1]. This approach focuses only on adulthood, not considering that the atherosclerotic process responsible for CV events suddenly begins after a certain age. Indeed, as proposed in 1990 by the “Ticking time” theory developed by the San Antonio Heart Study group, atherosclerosis would be the pivotal example of a disease whose process begins many years before its phenotype is fully expressed [2]. Hence, it is reasonable that the effect of CV risk factors does not initiate “magically” in adulthood, but that, in a cumulative albeit asymptomatic way, it contributes to vascular damage over the years. In support of this proposition, a brilliant autopsy study conducted on 94 individuals who were deceased at a young age from trauma showed that the severity of asymptomatic coronary and aortic atherosclerosis increases with the number of traditional CV risk factors [3]. Furthermore, in the last 20 years *a* growing literature has shown that the number of cardiovascular risk factors in youth is associated not only with the severity of subclinical coronary and aortic atherosclerosis in adulthood [4], but also with cardiovascular events and mortality [5].

Thus, the Committee on Preventing the Global Epidemic of Cardiovascular Disease recommended that interventions at “all stages of life course” should be performed in order to promote cardiovascular health by preventing acquisition and augmentation of risk. Early health education and early correction of an unfavorable lifestyle can prevent CV diseases in later life [6]. Indeed, it has been shown that the adoption of simple measures in youth, such as the conscious reading of food labels, is associated with a lower prevalence of CV risk factors, such as obesity and hypertension [7]. Furthermore, participation in afterschool programs on cardiovascular disease risk among youth with CV risk factors (e.g., severely obese young people) has been shown to be extremely effective in improving the cardiovascular risk profile [8]. Furthermore, where some psychosocial conditions of young people such as ethnical segregation worsen the cardiovascular risk profile, others such as belonging to certain religious affiliations (e.g., the Adventist Religion) would seem to mitigate it [9,10].

Data regarding cardiovascular risk profile among medical students are limited. Medical students may be a “protected” population of young people given the greater awareness they develop during medical school about what the CV risk factors are and how to address them. Thus, they have the tools to fall within a lower CV risk profile. Indeed, there is a well-established inverse relation between education and mortality from cardiovascular disease. This inverse gradient is mediated by a lower prevalence of modifiable CV lifestyle factors (such as smoking, inadequate physical activity, and diet) among educated subjects [11].

On the other hand, it is acknowledged that medical school is a long (six years in Italy) and demanding path that often limits students’ time and involves high stress, thus pushing students to have unhealthy lifestyle habits [12,13]. Accordingly, we conducted a cross-sectional survey among medical school students at the University of Milan, Italy to assess their CV traditional risk factors and global CV risk profile throughout the six years of school.

## 2. Materials and Methods

### 2.1. Questionnaire Administration and Recruitment

We performed a cross-sectional survey study through a peer-to-peer administration of an anonymous questionnaire (available in Appendix A) to all medical students attending the 3 distinct campuses (Centrale, San Paolo, Sacco/Vialba) of the University of Milan. A flyer (Figure 1) containing a quick response (QR) code was physically delivered to all medical students. By framing the QR code via smartphones, quick access to the online digital compilation of the questionnaire was possible. Since the academic course mandates compulsory attendance, during the first week of the 2019–2020 academic year (23–27 September) the flyers were placed on every student’s desk to ensure maximum coverage. To maximize the response rate, this procedure was repeated as “recall” one month later (24–25 October), and the questionnaire remained accessible from 23 September 2019 to 13 November 2019. The survey compilation implied informed consent to the use of the collected data under the guarantee of complete anonymity. Being a student-led initiative (Dr. Andrea Faggiano was still a student), approval from the ethics committee was not required. The average compilation time was 8 min. 

### 2.2. Questionnaire Characteristics

A single validated questionnaire was not adopted; instead, the UNIMI-HEART survey was created by the investigators using questions in accordance with the most recent guidelines regarding diet, lipid profile, blood pressure values, physical activity, alcohol intake, and cardiovascular prevention. The questionnaire was divided into five sections: (1) general data; (2) anthropometric, clinical, and lifestyle data; (3) diet; (4) pharmacological history; (5) perceived lifestyle variations during the six years of medical school (this section was not filled in by 1st year students). The basic demographic characteristics included age, gender, height, and weight. The body mass index (BMI) was automatically calculated in Microsoft Excel using the following formula: BMI = (Weight in kilograms) divided by (Height in meters squared). The BMI values of the students were used to categorize the subjects as underweight (BMI below 18.5), normal (BMI 18.5–24.9), overweight (25.0–29.9), or obese (30.0 plus). In addition to information regarding reported history of high blood pressure (BP), hypercholesterolemia, diabetes mellitus, and CV disease family history, data on smoking habits, alcohol and illicit drug use, sleep time, and physical activity were acquired and defined according to current guidelines. Specifically, European Society of Cardiology (ESC) guidelines were used to evaluate physical activity [1] and to set a total cholesterol cut-off (155 mg/dL) to identify low-risk subjects [14]. The number of alcoholic units per week (>10 if female or >15 if male) considered dangerous for cardiovascular health was defined by Canadian guidelines [15]. Because all data were collected using a survey, including the lipidic, glycemic, and blood-pressure profiles, all data were self-reported by the students and not measured by the investigators.

The 14-item PREDIMED SCORE, an index of Mediterranean diet adherence, was used to assess nutrition [16]. To limit the compilation time, validated questions regarding daily stress at work or home were posed [17]. Participants were asked to provide a self-evaluation regarding one’s lifestyle, adherence to the Mediterranean diet, and salt intake. Ideal cardiovascular health was evaluated according to the American Heart Association (AHA) definition [18]: the simultaneous presence of 4 ideal health behaviors (never smoked, 18.5 kg/m^2^ < BMI > 24.9 kg/m^2^, optimal physical activity, and optimal diet) and 3 ideal health factors (absence of dyslipidemia, hypertension, and diabetes).

### 2.3. Statistical Analysis

The database was formatted through Microsoft Excel software and subsequently imported into IBM-SPSS software ver. 25.0.2. The use of Stata software ver. 15.0 was also used for any comparisons or implementations of test output. For the statistical analyses, the continuously expressed variables were subjected to the Kolmogorov-Smirnov test to evaluate their normality, log-normality, or non-parametric distribution. The statistical investigation path began with the descriptive analysis of the variables and the related calculation of the main-position indices, or composition-ratio percentages. The inferential analyses, both parametric and non-parametric, were conducted using appropriate tests: Spearman’s r, χ squared and the exact Fisher test, Mann Whitney U-test, Kruskal-Wallis H-test. All results were tested at the α-significance level of 5%. The authors take full responsibility for the integrity of the data.

## 3. Results

Of the total 2700 students enrolled in the medical school at the University of Milan, 1183 (43.8% response rate) eventually completed the questionnaire. The distribution of responses over time shows that 3/4 were generally concentrated during the first week (23–27 September) while the rest were after the “recall” period of 24–25 October (Figure 2). The general characteristics of the students are listed in Figure 3. 

About 61% of the students were women, and the mean age of the sample was 22 years old. Being underweight (BMI ≤ 18.5 kg/m^2^) was more common among female students compared to male students, whereas being overweight was more common among male students compared to female students. About one-fifth of students had a history of smoking, with more than one out of ten students reporting being active smokers. Of these active smokers, around 90% smoked traditional tobacco and between 0 and 10 cigarettes per day. Furthermore, working students (25% vs. 12%, 95% CI 6.3985 to 20.9752; *P* = 0.001) and non-commuters (16% vs. 9%, 95% CI 3.1148 to 10.7322; *P* = 0.003) smoked considerably more. Surprisingly, the percentage of occasional illicit drug users was similar to the percentage of active smokers, with men more involved than women (19.4% vs. 10.3%, 95% CI 4.9483 to 13.4733; *P* = 0.001). 

About three-quarters of medical students were aware of their blood pressure (BP), with an increasing awareness as the medical-study years progressed (Central illustration, 3A). Of these, only a minority, (8%, 70 students) had BP values higher than optimal ones (120/80 mmHg), and these were more frequently men (16% vs. 4%; 95% CI 7.9279 to 16.5825; *P* = 0.001). Only one-third of students were aware of their lipid values, but awareness gradually increased to a peak of 60% during the 6th academic year (Central illustration, 3A). One-fifth of aware students (20%, 84) had total cholesterol values >155 mg/dL. Women more frequently had a lipid profile above normal levels (23.6% vs. 14.7%, 95% CI 0.9203 to 16.1644; *P* = 0.031).

Table 1 reports data on diet, physical activity, and sleep time. Regarding diet, few students had a PREDIMED score ≤5, denoting lack of adherence to the Mediterranean diet. Conversely, only 20% of students reported the adoption of an optimal diet, with a PREDIMED score ≥10, although most students believed they adopted the Mediterranean diet correctly. Moreover, half of the subjects believed in not to exceeding the limits of daily salt consumption, a third thought they do, and the rest had no idea [1]. Only 1 out 10 students reported not being stressed for university reasons, while around 4/10 were constantly stressed, especially women (43% vs. 28%, 95% CI 9.6634 to 20.3134; *P* = 0.001).

We measured the proportion of students having ideal cardiovascular health: only 22 students (1.9%) fell within this definition properly but, given the lack of some data regarding dyslipidemia and hypertension, it was estimated that the real presence in our population is 55 students (4.6%). This estimation was inferred through analyses made on the sample for which all the data were available (Appendix A available in the Appendix A). Moreover, as shown in Table 2, we found a rather important prevalence of CV risk factors among students. 

Furthermore, we had the chance to evaluate this prevalence at different phases of students’ medical school career. Of interest, the number of subjects with high BP grew as the academic course years progressed, as well as the number of subjects consuming excessive amounts of alcohol and the number of students constantly stressed for university reasons (Figure 4B–D). In addition, during the academic years more than half of individuals reported having reduced physical activity and three quarters having increased stress levels. Instead, the number of dyslipidemic subjects tended to decrease as the course years progressed, albeit only slightly. Similarly, about half of the subjects reported losing weight and improving their diet, although no improvement was observed in either PREDIMED or BMI.

## 4. Discussion

The main finding of this survey carried out in a large young Italian population, is a relatively high prevalence of traditional risk factors for atherosclerotic vascular disease in apparently healthy men and women in their third decade of life. Furthermore, these results were obtained from medical school students, in whom adequate knowledge of atherosclerosis mechanisms and strategies for prevention is expected. Of note, the CV risk profile worsened throughout the course of students’ medical school careers. Youth is a critical period: Not only is the correct awareness of individual risk exposure lacking [19], but it is also common to overestimate one’s own health status. Indeed, young people often believe there is still plenty of time to change their health behaviors and mitigate their own risk [20]. In reality, changing habits during this life period can be difficult. For example, a recent Framingham Offspring cohort study found that most young people with high lipid values tend to maintain them throughout their life [21]. Different from other similar studies, in our population some specific risk factors, such as smoking, were less prevalent. This finding is in line with the results of an Italian study [12] that found that 17% of medical students were active smokers, while a Croatian study found that smoker prevalence reached 30%, compared to the current prevalence of 13.5% [13]. Similarly, the proportion of observed overweight subjects (8%) is lower than that found in a study involving Serbian medical students (about 20%) [22]. On the other hand, the high prevalence of underweight, primarily among female students (18% vs. 3%, 95% CI 11.4540 to 18.0290; *P* = 0.001) is alarming, although only 4% of them are severely underweight (BMI < 16 kg/m^2^). It is worth mentioning that, especially among the population below 40 years old, underweight persons have a 2.3-fold greater risk of CV disease [23].

Beyond the low adherence to the Mediterranean diet, another interesting finding is the lack of insight regarding individual eating habits. In fact, even though 72% of students believed they adopted the Mediterranean diet correctly, only a quarter of them actually had a PREDIMED score >10. It is reasonable to expect similar data for salt intake: only 30% of students believed they use more salt than recommended while the literature reports that about 93% of young Italians exceed 5 g/day, a cut-off established by the World Health Organization (WHO) [24].

Moreover, although medical students have an awareness of their BP and lipid profile greater than that reported in similarly aged subjects [19], it is still unsatisfactory, especially regarding lipid values (35%). Unsurprisingly, awareness increases with the passing of the university years, as well as students’ knowledge of CV risk factors. Despite this, our study shows that important risk factors such as hypertension, sedentary lifestyle, and alcohol abuse tend to increase progressively. Recent studies have shown that those having risk factors such as smoking, hypertension, dyslipidemia, and diabetes in their youth are more prone to have subclinical atherosclerosis, as demonstrated by an increased carotid intima-media thickness and coronary artery calcification, proved by a high coronary artery calcification score, in later adulthood [4,25]. Additionally, the study conducted by Zhang et al. on 36,000 subjects showed that the cumulative young adult exposure to elevated BP and LDL-cholesterol was associated with an increased risk of coronary events and heart failure (HF) in later life, independently of later adult exposure. Furthermore, having systolic BP >120 mm Hg and diastolic BP >80 mmHg in young adulthood was associated with a 37% and 21% increased risk of HF, respectively [20]. In our study, a worrying peak of 13% of students with high BP was observed in the 5th year of the course (Central illustration, 2D), with 11.5% of them using antihypertensive drugs.

We hypothesize that the psychological and time commitments that medical school has a role in this worsening trend. The university-related stress levels are alarming, with about 40% of the students being constantly stressed and 75% they have seen their stress level increase over the years. Such a psychophysical condition may predispose medical students to inappropriate behaviors such as high alcohol consumption, substance abuse, and smoking. Indeed, a recent study showed that stress levels among students have been associated with higher blood pressure [26]. Not surprisingly, the smoking rate also increased in our study across the university years. The reduced time available for self-care inevitably leads to insufficient levels of physical activity, with about one-third of the students not practicing physical activity at all and most of them claiming to have become more sedentary throughout their university career. In support of these results, a recent study found that medical students begin medical school with better mental health indicators than age-similar college graduates in the general population, but that they develop higher rates of distress over time [27]. Indeed, many studies have demonstrated a high prevalence of burnout among medical students; for example Mazurkiewicz et al. showed that 71% of medical students at Mount Sinai School of Medicine in New York met criteria for burnout [28]. Therefore, our results support existing concerns that the learning environment and training process could contribute to the deterioration of mental and physical health in medical students. As proposed by Brazeu et al., medical student distress appears to be a “nurture” rather than a “nature” problem, indicating that changes in the learning environment are needed [27].

Finally, the overall risk profile of this population of medical students is far from ideal, even though more than 6 out of 10 subjects believe that they adopt a healthy lifestyle. In fact, only a few students (4.6%) can be considered as having ideal cardiovascular health. Approximately 5% of students, regardless of gender, have at least 3 of the 8 main cardiovascular risk factors: BMI > 24.9 kg/m^2^, active smoking, hypertension, dyslipidemia, diabetes, CV family history, inadequate diet (PREDIMED ≤ 5), and absence of physical activity. This rate likely identifies with high specificity those subjects who are at high risk of future CV events. On the other hand, if in addition to the 8 main risk factors, minor risk factors are also considered, such as BMI ≤ 18.5 kg/m^2^ [23], alcohol abuse [29], drug use [30], sleep time <6 h [31], up to 16% of students have at least 3 risk factors and only 28% of students have none. By also considering minor risk factors, it is possible to increase sensitivity in identifying possible subjects for future risk. Our data suggest that maximum attention should be paid to these subjects, stratifying their risk through a lifetime score, monitoring them carefully over time, implementing behavioral changes in diet and physical activity area, and if this is insufficient, implementing early introduction of pharmacological treatment.

### Limitations and Strengths

Our study has some limitations that should be highlighted. As with all surveys, the ascertainment of lifestyle habits and medical history is subject to recall bias and to concerns of accuracy with self-reported data. Although a 44% response rate may not seem high, we consider it a satisfactory result. In fact, the survey was conducted as an easy-to-do initiative by medical students to medical students without any traditional academic support. The study indeed was not promoted by the university nor sponsored or publicized by any of the professors. We suggest that a peer-to-peer anonymous approach that we used enhanced the honesty and reliability of the responses, especially those regarding subjects’ personal spheres, such as lifestyle and habits. The actual response rate may be higher because it is possible that we did not notify all 2700 students formally enrolled in the medical school program given any probable absences from face-to-face lessons at the moment the flyer was distributed. Importantly, general characteristics, such as gender and age, of the non-responders are similar to those of the responders. Non-responders, in fact, being enrolled in the same university, are part of the same academic courses, the same city, and a similar social context.

Our study is a cross-sectional description of different generations of students, so no clear conclusion can be made about the impact of medical education on each subject’s lifestyle and risk factors. We tried to mitigate this intrinsic limitation by including a specific section (fifth) of the questionnaire requiring students to self-evaluate the variations that they experienced during the course years. Only a prospective cohort study where medical students are evaluated at the beginning and at the end of their education may properly address this issue. Furthermore, by not having a control group, it is not possible to establish a causal link between the medical education path itself and the lifestyle changes and behaviors we observed among our students.

The main strengths of this study include the large sample size and the innovative method of conducting the survey. We emphasize the practicality, convenience, and speed of data collection through the online questionnaire accessible via QR code. It may be that an eye-catching flyer with a “mysterious” QR code encouraged students, otherwise generally hesitant to complete surveys if not obliged, to fill in the questionnaire. To our knowledge, in the current medical literature there are only a few studies based on QR code surveys. We believe that this methodology can be widely used in the future to create reliable survey databases, especially in the COVID-19 era in which social interactions must be limited as much as possible [32]. Furthermore, our study provides new data to the current limited literature regarding medical students’ health status and lifestyle.

## 5. Conclusions

In conclusion, the data from the UNIMI HEART survey remind us that the time has probably come to shift the gaze of primary prevention from adults to young people. Furthermore, the present study suggests that even a population considered “protected”, such as that of young medical students, can show an unsatisfactory cardiovascular risk profile, and that medical school itself, being demanding, long, and stressful, could have a role in worsening the lifestyle. To clarify this aspect, a prospective study is needed, and it would be desirable to conduct this study in other faculties (e.g., economics, law, etc.). It seems like an appropriate time to question the medical education system, which trains its students without guaranteeing them the correct lifestyle it teaches. Such training issues also can affect patients: Can doctors in clinical practice properly educate their patients if they cannot apply the knowledge to themselves?

## Figures and Tables

**Figure 1 jcm-10-01343-f001:**
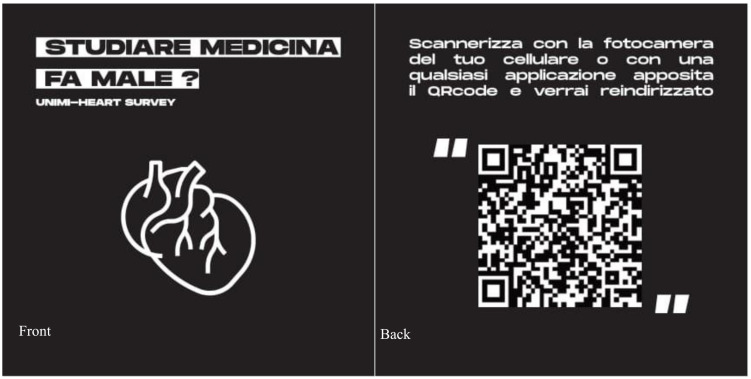
QR code flyer. Front and back of QR code flyer delivered to the students.

**Figure 2 jcm-10-01343-f002:**
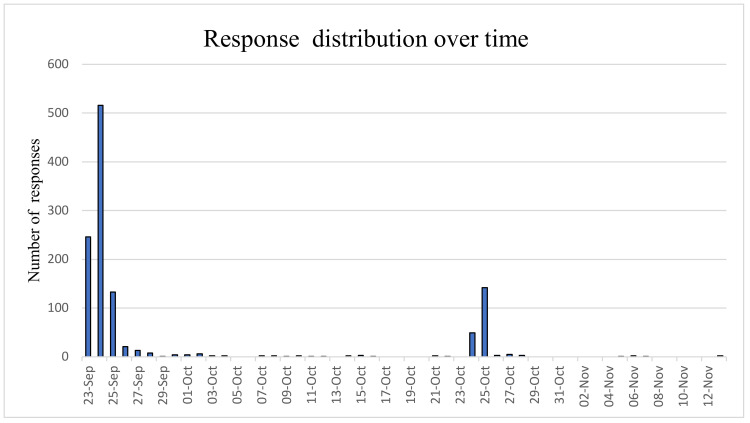
Distribution of responses over time: 1° peak during the academic year’s first week and 2° peak after the “recall” flyers delivery.

**Figure 3 jcm-10-01343-f003:**
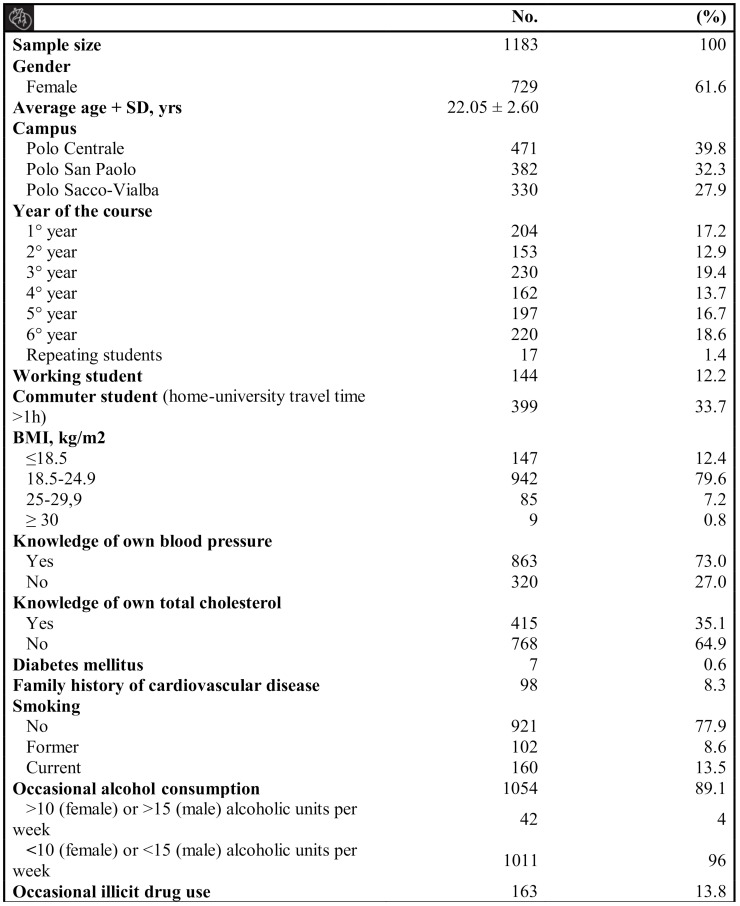
Students’ general and clinical characteristics.

**Figure 4 jcm-10-01343-f004:**
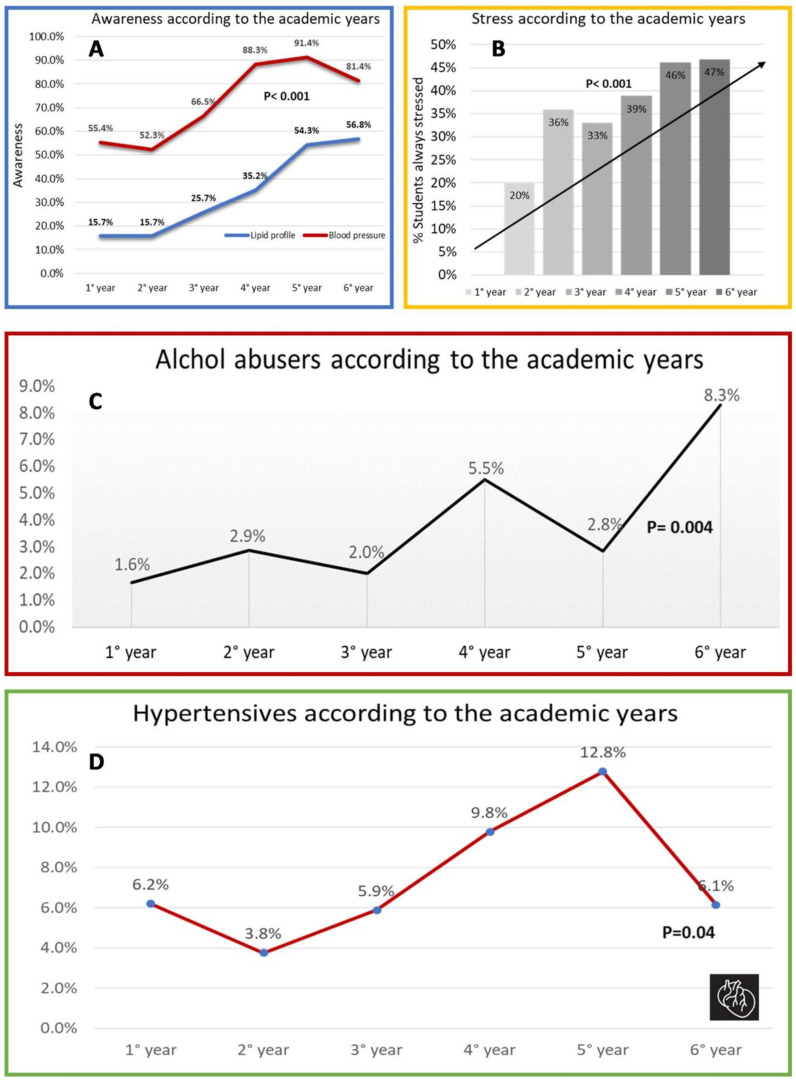
Student’s awareness (**A**), stress (**B**), alcohol abuse (**C**), and high blood pressure (**D**) according to academic years. (**A**) Blood pressure and lipid profile awareness increase with the academic course years. (**B**) University-related stress progressively rises over the course of academic years. (**C**) Alcohol abusers (>10 (female)/>15 (male) alcoholic units per week) according to the course years. (**D**) Number of hypertensive students grows over the years of the course.

**Table 1 jcm-10-01343-t001:** Physical activity, sleep time, and adherence to the Mediterranean diet (indicated by PREDIMED SCORE) among students. Light activity: the heart beats just faster than normal, while exercising it is possible to speak without any problems. Moderate activity: the heart beats faster than normal, while exercising it is possible to speak but with a little trouble. Intense activity: the heart beats much faster than normal, while exercising it is difficult to speak. The “No physical activity” category includes students who do not engage in physical activity at all or practice light physical activity but not every week. The “intermediate” category includes students who practice light activity every week or moderate activity <30 min/day or <5 times/week or intense activity <20 min/day or <3 times/week. Finally, the “optimal” category includes students who practice moderate activity >30 min/day at least 5 times/week or intense activity >20 min/day at least 3 times/week.

PREDIMED Score	No.	(%)
Inadequate diet: score ≤ 5	121	10.3
Intermediate diet: score 6–9	831	70.2
Optimal diet: score ≥ 10	231	19.5
Physical activity		
No physical activity: Never Light activity not every week	387	32.7
Intermediate physical activity: Light activity every week Moderate activity <30 min/day or <5 times/week Intense activity <20 min/day or <3 times/week	511	43.2
Optimal physical activity: Moderate activity >30 min/day at least 5 times/week Intense activity >20 min/day at least 3 times/week	285	24
Sleep time		
<6 h	190	16.1
6–9 h	984	83.2
>9 h	9	0.8

**Table 2 jcm-10-01343-t002:** Presence of CV risk factors among students. ¹ BMI > 24.9 kg/m^2^, active smoking, hypertension, dyslipidemia, diabetes, CV family history, inadequate diet and inadequate physical activity. ^2^ Eight main CV risk factors + BMI ≤ 18.5 kg/m^2^, alcohol abuse, drug use, sleep time <6 h or >9 h.

Number of 8 Main ¹ CV Risk Factors	No. (%)	Number of 12 Extended ^2^ CV Risk Factors	No. (%)
0	495 (41.8%)	0	333 (28.1%)
1	432 (36.5%)	1	403 (34.1%)
2	198 (16.7%)	2	260 (21.9)
≥3	58 (4.9%)	≥3	187 (15.9%)
Total	1183 (100%)	Total	1183 (100%)

## Data Availability

The data underlying this article will be shared on reasonable request to the corresponding author.

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
