# Peer review of "Global Cardiovascular Risk Profile of Italian Medical Students Assessed by a QR Code Survey. Data from UNIMI HEART SURVEY: Does Studying Medicine Hurt?"

_jcm, 2021, doi:10.3390/jcm10071343_

Round 1
Reviewer 1 Report
As the system has not been able to tranfer my reviewer comments, I will send these per email

Author Response
Point 1: Introduction, general: there may be more data on general risk behaviour and risk patterns of the youth supported or attenuated by special beliefs, education and psychosocial conditions– please include.
Response 1: Thank you reviewer for the advice. We have implemented the introduction as follow:
“Thus, the Committee on Preventing the Global Epidemic of Cardiovascular Disease recommended that interventions at 'all stages of life course' should be performed in order to promote cardiovascular health by preventing acquisition and augmentation of risk. Early health-education and early correction of an unfavorable lifestyle can prevent CV diseases in later life. Indeed, It has been shown that the adoption of simple measures in youth, such as the conscious reading of food labels, is associated with a lower prevalence of CV risk factors, such as obesity and hypertension. Furthermore, participation in afterschool programs on cardiovascular disease risk among youth with CV risk factors, (e.g. severely obese young people), has been shown to be extremely effective in improving the cardiovascular risk profile. Furthermore, where some psychosocial conditions of young people such as ethnical segregation worsen the cardiovascular risk profile, others such as belonging to certain religious affiliations (e.g. the Adventist Religion) would seem to mitigate it.”
Point 2: Introduction, the role of education at school should be mentioned.
Response 2: Thank you Reviewer for the suggestion. To better explain the impact of school education on cardiovascular disease we have added this sentence in the introduction:
“Indeed, there is a well-established inverse relation between education and mortality from cardiovascular disease. This inverse gradient is mediated by a lower prevalence of modifiable CV lifestyle factors (such as smoking, inadequate physical activity and dietary) among educated subjects.”
Point 3: Please check whether there are more recent data on the time course of atherosclerosis during live time.
Response 3: Thank you reviewer. We have implemented the introduction with some recent data regarding the development of atherosclerosis since youth to adulthood. The following sentence has been added:
“Furthermore, in the last 20 years a growing literature has shown that the number of cardiovascular risk factors in youth is associated not only with the severity of subclinical coronary and aortic atherosclerosis in adulthood, but also with cardiovascular events and mortality.”
Point 4: Methods and results Table 1: It is remarkable, that any participating person was able to categorize the individual BMI – I think this only can be a rough estimate.
Response 4: Thank you very much for raising this point. We haven't been clear enough about this in the methods section; the students provided weight and height and automatically the excel worksheet calculated the body mass index (BMI) of each student. Once we have obtained all the BMIs we have proceeded to categorize the students as shown in Table 1.
To be clearer we have made the following change in the manuscript:
The sentence "The basic demographic characteristics included age, gender, height, weight and body mass index (BMI)" was changed to "The basic demographic characteristics included age, gender, height and weight. The body mass index was then automatically calculated from Microsoft-Excel software using the following formula: BMI = (Weight in kilograms) divided by (Height in meters squared).The BMI values of the students were used to categorize the subjects as underweight (BMI below 18.5), normal (BMI 18.5 - 24.9), overweight (25.0 - 29.9), or obese (30.0 plus).
Point 5: Methods: the “online” Figure 2 (supplements) is not readily clear for the reader and should be explained in detail within the figure legend.
Response 5: Thank you Reviewer. We have implemented the figure legend with the following sentences:
“The Ideal Cardiovascular health is the simultaneous presence of 4 ideal health behaviors (never smoked, 18.5 kg / m2 24.9 kg / m2, optimal physical activity and optimal diet) and 3 ideal health factors (absence of dyslipidemia, hypertension and diabetes). Not having available the lipid and blood pressure profile of all the students, an estimate was made based on the available data. Only 61/1183 students had the simultaneous presence of the 5 behaviors and factors available to everyone (BMI, diabetes, smoking status, diet, physical activity). Therefore we had to understand how many of these 61 subjects also had a normal lipid and blood pressure profile. Of these 61, we had lipid and blood pressure data of 27 students: 2/61 were dyslipidemic while 1/61 was hypertensive; therefore only 24/27 students had the ideal cardiovascular health. We did not have available neither the lipid value nor the blood pressure of 12/61 subjects, therefore the estimate of how many were neither dyslipidemic nor hypertensive is based on the ratio 24/27 (obtained from the subjects we had all the data). 21/61 patients had normal blood pressure but unknown lipid values, therefore the estimate of dyslipidemic subjects among them was carried out using the ratio 24/26 (proportion of students with known normal lipid values). Finally, a student was not dyslipidemic but blood pressure values were unknown: to estimate if his lipid values were normal we used the ratio 24/25 (proportion of known normotensive students). Overall, only 55/1183 students (4.65%) had the ideal cardiovascular health.”
Point 6: Results: The supplemental figure 1, showing the timeline of responses to the questionnaire, should be shown within the main text.
Response 6: Thank you very much for the advice. We have added the figure within the main text.
Point 7: Results, table 2: the categorization of “no physical activity”, “moderate…. and optimal physical activity” needs to be explained in more detail in the figure legend, as each category includes 2-3 sub-categories. Actually this may be somewhat confusing.
Response 7: Thank you Reviewer. We have implemented the figure legend as follow:
“Light activity: the heart beats just faster than normal, while exercising it is possible to speak without any problems. Moderate activity: the heart beats faster than normal, while exercising it is possible to speak but with a little trouble. Intense activity: the heart beats much faster than normal, while exercising it is difficult to speak. The "no physical activity" category includes students who do not engage in physical activity at all or practice light physical activity not every week. The "intermediate" category includes students who practice light activity every week or moderate activity <30 min / day or <5 times / week or intense activity <20 min / day or <3 times / week. Finally, the "optimal" category includes students who practice moderate activity> 30 min / day at least 5 times / week or intense activity> 20 min / day at least 3 times / week”.
The references are in the main text

Reviewer 2 Report
-better specify the validation of the questionnaires used
-in the material and methos secton better specify the collection of lipid profile, glycemia and blood pressure (not clear if measured or reported)
The main limitaton is the lack of a control group to answer the question posed by authors in the title> Does studying medicine hurt?there is no control group to understand if the changes are due to university study. I suggest to reformule the title or better discuss this point in the discussion section
Author Response
Point 1: better specify the validation of the questionnaires used.
Response 5: Thank you very much Reviewer. We have added this sentence in the methods section
“A single validated questionnaire was not adopted, instead the UNIMI-HEART survey was created by the investigators using questions in accordance with the most recent guidelines regarding diet, lipid profile, blood pressure values, physical activity, alcohol intake and cardiovascular prevention”.
Point 2: In the material and methods section better specify the collection of lipid profile, glycemia and blood pressure (not clear if measured or reported).
Response 2: Thank you very much for the advice. We have implemented the method section with the following sentence:
“Being a survey, all the data, including the lipid, glycemia and blood pressure profiles were self-reported by the students and not measured by the investigators.”
Point 3:The main limitaton is the lack of a control group to answer the question posed by authors in the title> Does studying medicine hurt?there is no control group to understand if the changes are due to university study. I suggest to reformule the title or better discuss this point in the discussion section.
Response 3: Thank you very much for having raised this point.
Although the title is not totally data-driven, we would prefer to keep it in order to arouse interest and stimulate a constructive discussion about the topic. As you advised we have underlined this important limitation in the discussion as follows
“Furthermore, not having used a control arm, it is not possible to state that the medical education path is actually responsible for the lifestyle changes and behaviors we encountered among our students”
Furthermore, to better underline the possible relationship between medical education and mental and physical distress, we added the following two paragraphs in the discussion:
“Indeed, recently stress levels among students have been associated with higher levels of blood pressure”
“In support of our hypothesis, a recent study found that medical students begin medical school with better mental health indicators than age-similar college graduates in the general population but that they develop over the years higher rates of distress. Indeed, many studies have demonstrated high prevalence of burnout among medical students; for example Mazurkiewicz et al showed that 71% of medical students at Mount Sinai School of Medicine in New York met criteria for burnout. Therefore, our results support existing concerns that the learning environment and training process could contribute to the deterioration of mental and physical health in medical students. As proposed by Brazeu et al, medical student distress appears to be a “nurture” rather than a “nature” problem, indicating that changes in the learning environment are needed”
The references are in the main text.

Round 2
Reviewer 2 Report
no further comments